# Principles of Rapid Weight Loss in Female Sambo Athletes

**DOI:** 10.3390/ijerph182111356

**Published:** 2021-10-28

**Authors:** Nikola Todorović, Marijana Ranisavljev, Borislav Tapavički, Andrea Zubnar, Jovan Kuzmanović, Valdemar Štajer, Damir Sekulić, Šime Veršić, Sergey Tabakov, Patrik Drid

**Affiliations:** 1Faculty of Sport and Physical Education, University of Novi Sad, 21000 Novi Sad, Serbia; nikolatodorovic1708@gmail.com (N.T.); marijanaranisavljev@gmail.com (M.R.); kuzman_07@hotmail.com (J.K.); stajervaldemar@yahoo.com (V.Š.); 2Department of Physiology, Faculty of Medicine, University of Novi Sad, 21000 Novi Sad, Serbia; borislav.tapavicki@mf.uns.ac.rs (B.T.); andrea.zubnar@mf.uns.ac.rs (A.Z.); 3Faculty of Kinesiology, University of Split, 21000 Split, Croatia; dado@kifst.hr (D.S.); sime.versic@kifst.hr (Š.V.); 4Russian State University of Physical Education, Sports, Youth and Tourism, 105122 Moscow, Russia; samboskif@mail.ru

**Keywords:** rapid weight loss, weight class, weight control, combat sport, sambo

## Abstract

In combat sports, competitors are separated into weight divisions, to create equality between the athletes. Consequently, rapid weight loss (RWL) is a common practice in combat sports. Although the methods used to induce RWL are similar among athletes, currently, there are limited data concerning the RWL methodologies used by sambo athletes. Therefore, this study aimed at determining RWL procedures among female sambo athletes. Participants in the study were top-level athletes competing at the World Sambo Championship held in Novi Sad. A total sample of 47 female sambo athletes, of whom 24 were seniors (27.3 ± 4 year/age, 1.61 ± 0.09 m/height, 61.8 ± 8.87 kg/weight) and 23 juniors (18.7 ± 0.8 year/age, 1.66 ± 0.07 m/height, 63.7 ± 12.1 kg/weight), were examined in the study. To determine RWL methods, data were collected through a standardized questionnaire. As a result, 88.7% of the female sambo athletes declared that they intentionally cut their weight before the competition. The most commonly used methods were gradual dieting, followed by sauna, fluid restriction, and skipping meals. The most considerable influence on the RWL strategies of athletes came from coaches and personal trainers, while physicians and dietitians were far less influential. The results obtained from this representative sample highlight the most common practices concerning weight cutting prior to competition among females. Therefore, there is a need to inform and educate both athletes and coaches about the potentially harmful effects of RWL in combat sports.

## 1. Introduction

Sambo is a combat sport developed in the Soviet Union at the beginning of the 20th century. It was created as a training tool and used by the Soviet Red Army to improve their combat abilities and physical fitness [1]. After 80 years of existence and development, Sambo received attention and recognition from the International Olympic Committee (IOC) and made the first steps toward inclusion into the Olympic games [2]. Weight categories have been used for centuries to equalize competition in various different sports, especially in sports for which the competitors’ physical strength is perceived as a crucial factor in their ultimate success. In combat sports, competitors are separated into weight divisions in an attempt to create equality between the athletes [3]. Athletes try to cut (reduce) their weight because of competitive and tactical advantages or psychological reasons. This also applies to sambo athletes. Earlier studies revealed that nearly 90% of male and female judo athletes engage in rapid weight loss (RWL) before competitions [4]. Similar results were observed in other combat sports [5,6]. Based on previous research, many athletes attempt to achieve their target weight via a combination of harmful methods, including severe energy restriction and dehydration [7,8]. Regardless of the type of combat sport, the methods of inducing RWL are very similar, often starting by reduced ingestion of fluids, fasting, and skipping meals, combined with supplementation, high training loads, plastic suit training, and sauna use [9,10]. Coming from a perspective that prioritizes athletes’ health and safety, it is important to outline that RWL practices are not without consequences, primarily affecting the increase of certain blood biomarkers and psychological parameters, such as tension, anxiety, and depression [11]. Another important topic that must be addressed is the cognitive health of athletes. Several mechanisms explain why cognitive and mental health may be altered among athletes who implement RWL prior to competitions, including dehydration, hypoglycemia, and the impact of affective changes during RWL on cognitive/mental health [12].

This article pays close attention to acute weight reduction, which is widely discussed in the literature [3,13,14]. However, there are limited data concerning RWL among sambo athletes. To date, and to the best of the authors’ knowledge, only two studies have examined the magnitude of RWL practices among sambo athletes [15,16]. This study used a similar database to the two previously mentioned studies, but examined different samples. In the first study, researchers evaluated the overall status of RWL methods, combining results from male and female sambo athletes. In contrast, the differences between senior and junior male sambo athletes were evaluated in the second study. No previous study has been conducted on female athletes. Consequently, this study aims to identify the methodologies of RWL, with the main hypothesis set to define the most widespread methods and sources of influence used by female sambo athletes and to determine any differences in RWL strategies among senior and junior sambo athletes.

## 2. Materials and Methods

### 2.1. Participants

Participants in the study were top-level athletes competing at the World Sambo Championship held in Novi Sad. A total of 483 athletes from 34 countries were competing in the tournament. Of these, 199 participants completed the questionnaires. However, 146 athletes were excluded from the study because they were either male athletes or female athletes younger than 18 years of age. In a further analysis, only female sambo athletes (18+ years of age) who answered ‘yes’ on the specific question ‘Have you ever lost weight in order to compete?’ were included. A total sample of 47 athletes from 12 countries were included for statistical analysis. They were further divided into senior (27.3 ± 4 year/age, 1.61 ± 0.09 m/height,61.8 ± 8.87 kg/weight) and junior groups (18.7 ± 0.8 year/age, 1.66 ± 0.07 m/height, 63.7 ± 12.1 kg/weight). A detailed flow chart of participants is shown in Figure 1. The study obtained ethical approval from the ethical committee of the University of Novi Sad, Serbia (Ref. No. 46-06-02/2020-1) and was conducted according to the Helsinki Declaration. Furthermore, all sambo athletes gave written informed consent, with complete information about the study and questionnaires provided by the investigator.

### 2.2. Data Assessment

To assess the RWL methods among female sambo athletes, we adopted a standardized RWL questionnaire developed by Artioli et al. [17]. The questionnaire consists of questions relating to RWL behaviors, including the sources of influence and the methods used to cut weight before a competition, but also relating to personal information, nutrition status, and weight loss history. The frequency at which these strategies were adopted for weight loss strategy methods was classified as (1) always, (2) sometimes, (3) almost never, (4) never used, and (5) not used anymore; where only 1 and 2 were considered as ‘adopted’. Furthermore, concerning the sources of influence, the frequencies were reported as (1) not influential, (2) little influential (3), unsure, (4) somewhat influential, and (5) very influential; where only 4 and 5 were considered as a source of influence. In addition, the questionnaires were translated from the original Portuguese language to several languages (e.g., Russian, French, Serbian, and Spanish) to facilitate data collection. The questionnaires were filled out in the competition venue. The investigators provided detailed information about the questionnaires and answered any participant inquiries during the procedure.

### 2.3. Statistical Analysis

All statistical and data analyses were conducted using the SPSS statistical software ver. 24.0 (IBM SPSS Statistics, Chicago, IL, USA). First, data were checked for normality using a Kolmogorov-Smirnov test. Then we calculated descriptive statistics for variables of height, weight, age, and experience. Differences between seniors and juniors were evaluated using a *t*-test, with the significance level set at *p* ≤ 0.05. For determining the differences in RWL techniques and sources of influence, a chi-square crosstabulations test was conducted.

## 3. Results

A total of 88.7% (*n* = 47) of female sambo athletes declared that they intentionally cut their weight before competitions. It was reported that they cut approximately 3.5 kg, starting on average around 12 days before the competition. In addition, they reported that they had begun these kinds of RWL practices at 15 years of age. There were no statistical differences between these two female groups of athletes in their practice of RWL (see Table 1).

The greatest influence on the RWL strategies of athletes (calculated as the sum of the answers ‘somehow influential’ and ‘very influential’) came from coaches (67.4%) and personal trainers (43.7%), while physicians (12.7%) and dietitians (19.1%) had a smaller influence. The only statistically significant difference between subgroups was noted for the influence of fellow sambists (25% vs. 21.7%, *p* = 0.02) (see detailed in Table 2).

Concerning the methods that athletes applied during RWL, there were no statistical differences between seniors and juniors. The most commonly used methods by both groups combined for RWL (calculated as the sum of the answers ‘always’ and ‘sometimes’) were gradual dieting (83%), followed by sauna (72.9%), fluid restriction (72.4%), and skipping meals (70.3%). The less common methods were the usage of laxatives (6.4%), vomiting (6.4%), diuretics (12.7%), and diet pills (10.7%) (see detailed in Table 3).

## 4. Discussion

The goal of this study was to determine the methods and influences among female sambo athletes during RWL practices. This study is one of the first studies to evaluate the issue of RWL among elite female athletes. It is essential to put a focus on this specific problem, in terms of their well-being and long-term health. Almost all participants in this study, both seniors and juniors, reported cutting weight before competitions (88.7%). There were only small differences between young and experienced athletes, and this occurrence could partly be explained by the fact that most were influenced by coaches. In addition, sambo athletes start their RWL strategies at a very young age, which could also have contributed to similar results being obtained between seniors and juniors. Furthermore, it was detected that senior athletes seek advice more often from their fellow sambists and share their experiences among themselves concerning RWL methods. The most significant influence on the RWL strategies of both seniors and juniors was had by coaches and personal trainers. Another interesting point, although not statistically significant, was the percentage of body weight that athletes regain after RWL. It was noticed that seniors regain much more weight in comparison to juniors (5.2 kg vs. 2.9 kg, respectively). This jump or yo-yo effect is not very rare or uncommon. In previous research, it was also observed that weight usually rebounds during the gaining phase to a value higher than the starting value [18,19]. This causes athletes to conduct more aggressive methods in the future to enter the desired weight category. Moreover, commonly used RWL could lead to both acute (lower energy state) and chronic consequences (development of cardiovascular diseases) [20]. Similar findings were noted among male sambists, where senior athletes also regained more weight compared to juniors [16]. Furthermore, in the study by Drid et al. [15], conducted with male and female sambo athletes, complementary results were obtained in terms of the methods and influences used during RWL. The most common methods in our study were gradual dieting and skipping meals, fluid restriction, and sauna use. In the study conducted on Judo athletes by Artioli et al. [4], similar methods were reported. In addition, comparable results were found among high-school wrestlers [8,21]. Coaches and personal trainers were the most influential among our sample. In alignment with our findings, other studies reported that the coach is the most influential person when athletes intend to cut their weight [22]. Educating trainers about the potential side effects of weight loss should be prioritized in the future. Furthermore, we noticed that dietitians and physicians had low levels of influence among athletes. It seems that there is a certain gap of trust between science and practice. Practitioners must prioritize athletes’ health and outline safety considerations concerning RWL. A possible practical implication could be holding more educational seminars concerning the topic of RWL. The main topic of education should be the emphasis on the harmfulness of excessive weight loss, especially if it exceeds 5% of body weight [23], and to highlight the harmfulness of certain methods regarding the long-term health of athletes.

RWL can negatively affect cognitive performance. Numerous mechanisms have been suggested in an attempt to explain why cognitive changes occur during RWL. These explanations include the systematic effects of dehydration, hyperglycemia, and affective changes to memory and cognitive perception [11,24,25]. For example, in the study of Choma et al. [26], wrestlers achieved a poorer cognitive score on two out of six tests and had negative mood changes after RWL, compared to the control group. Marriott and Carlson [27] found similar results. These changes could possibly negatively affect performance during competition, but most importantly, they can have negative implications for the long-term health-related quality of life among athletes. Furthermore, RWL can induce both acute [28,29] and chronic [30] hormonal imbalances. This could be followed by loss in bone density and impair the immune function of athletes [30,31]. Finally, RWL can significantly increase the risk of injuries [32]. In addition, it was noted that females had a greater injury risk than males, when conducting RWL strategies. All of these findings lead to an important question: Is it time to ban RWL from combat sports? In one recent opinion paper, Artioli [33] suggested that RWL should be banned in combat sports. In his detailed explanation, he stated that all three world anti-doping agency criteria to ban a substance or method are met. Thus, RWL could, indeed, enhance sport performance, have certain risks to athletes’ health, and, although debatable, violate the spirit of the sport. Banning RWL procedures would prioritize the health and safety of the athlete, fair play, and finally would improve or bring back the true spirit of the sport.

The current study has a few limitations. First, the methodology is lacked an experimental approach. Due to COVID-19, the championship was organized following a particular set of procedures named ‘the bubble’; a sort of quarantined sports competition, whereby athletes were only able to spend time in a hotel or sports arena, and researchers were unable to evaluate any other physiological parameters of RWL, such as detailed body composition parameters (e.g., bioimpedance), blood biomarkers (e.g., CK, Myoglobin, Aldolase, LDH), or urine samples (e.g., Urca, BUN). For future investigations, the authors may consider the use of the bioimpedance methods, important for determination of changes regarding body weight and cellular health [34]. In addition, we did not track any changes before and after RWL procedures. Second, the current study only had an explorative purpose, and even though we ensured that the questionnaire was anonymous, the participants could have been biased in their answers, and our investigation could not reveal if the athletes were sincere regarding their adopted practices. Despite these limitations, the study covers a topic that has never been previously addressed in female sambo athletes and provides a foundation for future studies.

## 5. Conclusions

In this cross-sectional study, we observed methods and influence on RWL among elite female sambo athletes. The results obtained from this representative sample highlight the most common practices concerning weight cutting prior to competition. The majority of athletes (88.7%) practice RWL strategies. We did not find any differences between younger and older athletes, most probably because athletes start RWL practices at a very young age. The most widely used methods (dehydration, restricted diet, etc.) can be harmful and negatively affect athletes’ long-term health. Therefore, RWL, whether in the male or female population, is a major problem. In addition to the practices used by athletes in this paper, we indicated in detail the impact of RWL on several aspects of health. Both of the groups reported that the most influential person on their RWL strategies was their coach, while at the same time, experienced athletes sought advice more often from their fellow sambists. It is concerning that athletes rarely look for advice from a dietician or a physician. Finally, with all of the above in mind, athletes and coaches need to find better ways of practicing RWL or eventually even consider stopping its use. The education of coaches and athletes could be key to the mission of improving and advancing sambo.

## Figures and Tables

**Figure 1 ijerph-18-11356-f001:**
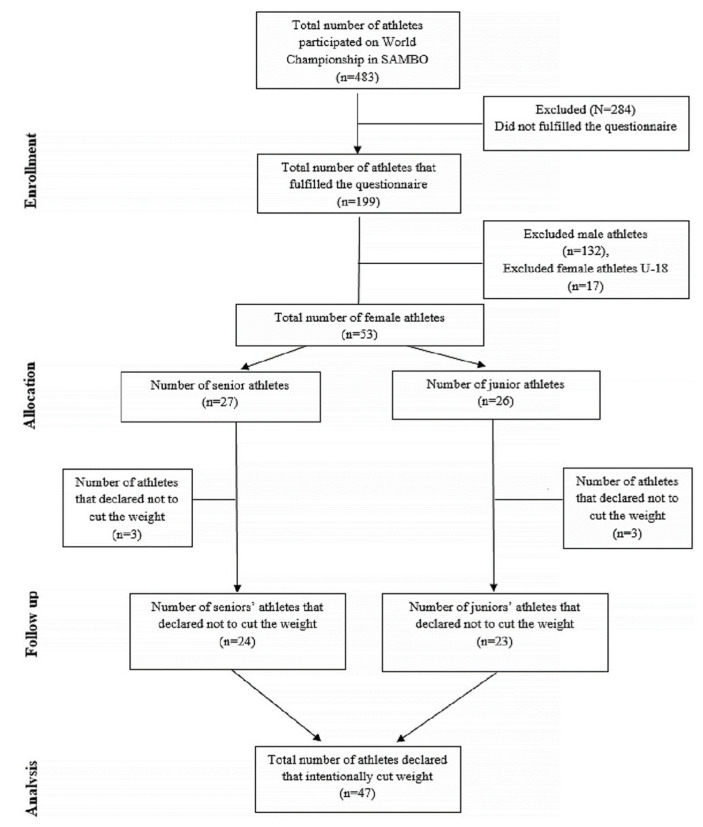
Flow chart of subject selection.

**Table 1 ijerph-18-11356-t001:** Weight reduction history reported by two categories of sambo competitors.

	Group	Mean	SD	*t*	*p*
How much weight do you usually cut before the competition? (kg)	Senior	3.5	1.4	1.26	0.22
	Junior	3	1.5		
How many days before the competition do you usually cut weight? (days)	Senior	11.9	10.2	0.17	0.97
	Junior	11.87	9		
At what age did you start to cut weight before a competition? (years)	Senior	15.5	5.2	1.26	0.27
	Junior	14.2	2.7		
How much weight do you usually regain after a competition? (kg)	Senior	5.2	9.7	1.24	0.22
	Junior	2.7	1.3		

Legend: SD—standard deviation; *t*—independent *t*-test; *p*—probability.

**Table 2 ijerph-18-11356-t002:** Frequency distribution of athlete influential persons during RWL.

Source of Influence	Group	Not Influential	Little Influential	Unsure	Somehow Influential	Very Influential	χ2	*p*
Teammate	Seniors	9 (37.5%)	6 (25%)	0 (0%)	6 (25%)	3 (12.5%)	2.75	0.6
	Juniors	6 (26.1%)	7 (30.4%)	2 (8.7%)	5 (21.7%)	3 (13%)		
Fellow sambist	Seniors	13 (54.2%)	0 (0%)	5 (20.8%)	4 (16.7%)	2 (8.3%)	11.5	0.02 *
	Juniors	9 (39.1%)	8 (34.8%)	1 (4.3%)	3 (13%)	2 (8.7%)		
Physician	Seniors	12 (50%)	8 (33.3%)	0 (0%)	1 (4.2%)	3 (12.5%)	6.03	0.3
	Juniors	16 (69.6%)	3 (13%)	1 (4.3%)	0 (0%)	2 (8.7%)		
Personal trainer	Seniors	10 (41.7%)	3 (12.5%)	0 (0%)	8 (33.3%)	5 (20.8%)	6.4	0.27
	Juniors	8 (34.8%)	3 (13%)	3 (13%)	5 (21.7%)	3 (13%)		
Coach	Seniors	3 (12.5%)	2 (8.3%)	0 (0%)	11 (45.8%)	7 (29.2%)	5.02	0.29
	Juniors	5 (21.7%)	3 (13%)	2 (8.7%)	5 (21.7%)	8 (34.8%)		
Parents	Seniors	9 (37.5%)	3 (12.5%)	2 (8.3%)	4 (16.7%)	5 (20.8%)	2.17	0.83
	Juniors	7 (30.4%)	5 (21.7%)	1 (4.3%)	5 (21.7%)	5 (21.7%)		
Dietitian	Seniors	15 (62.5%)	2 (8.3%)	2 (8.3%)	2 (8.3%)	3 (12.5%)	3.31	0.51
	Juniors	17 (73.9%)	0 (0%)	2 (8.7%)	3 (13%)	1 (4.3%)		

Legend: χ2—chi-square test; *p*—probability; *—significant.

**Table 3 ijerph-18-11356-t003:** Frequency distribution of methods used during RWL.

Source of Influence	Group	Always	Sometimes	Rarely	Never	Do not Use It Anymore	χ2	*p*
Gradual dieting	Seniors	10 (41.7%)	10 (41.7%)	4 (16.7)	0 (0%)	0 (0%)	2.7	0.44
	Juniors	10 (43.5%)	9 (39.1%)	2 (8.7%)	2 (8.7%)	0 (0%)		
Skipping meals	Seniors	5 (20.8%)	12 (50)	3 (12.5)	3 (12.5)	1 (4.0)	2.42	0.66
	Juniors	6 (26.1%)	14 (60.9%)	2 (8.7%)	1 (4.3%)	0 (0%)		
Fasting	Seniors	2 (8.3%)	12 (50%)	1(4.2%)	6 (25%)	3 (12.5%)	0.95	0.92
	Juniors	2 (8.7%)	13(56.5%)	2 (8.7%)	4 (17.4%)	2 (8.7%)		
Restricting fluid ingestion	Seniors	7 (29.2%)	8 (33.3%)	3 (12.5%)	4 (16.7%)	2 (8.3%)	3.52	0.47
	Juniors	6 (26.1%)	12 (52.2%)	3 (13%)	2 (8.7%)	0 (0%)		
Increased exercise	Seniors	10 (41.7%)	7 (29.2%)	6 (25%)	0 (0%)	1 (4.2%)	0.54	0.91
	Juniors	8 (34.8%)	7 (30.4%)	6 (26.1%)	0 (0%)	2 (8.7%)		
Training in heated room	Seniors	4 (16.7%)	7 (29.2%)	7 (29.2%)	6 (25%)	0 (0%)	3.63	0.30
	Juniors	4 (17.4%)	12 (52.2%)	5 (21.7%)	2 (8.7%)	0 (0%)		
Sauna	Seniors	7 (29.2%)	8 (33.3%)	1 (4.2%)	5 (20.8%)	3 (12.5%)	5.4	0.25
	Juniors	7 (30.4%)	12 (52.2%)	2 (8.7%)	2 (8.7%)	0 (0%)		
Training in plastic suits	Seniors	1 (4.2%)	12 (50%)	3 (12.5%)	5 (20.8%)	3 (12.5%)	8.57	0.07
	Juniors	7 (30.4%)	8 (34.8%)	2 (8.7%)	6 (26.1%)	0 (0%)		
Using a plastic suit all day	Seniors	2 (8.3%)	5 (20.8%)	3 (12.5%)	11 (45.8%)	3 (12.5%)	1.07	0.9
	Juniors	2 (8.7%)	6 (26.1%)	3 (13%)	11 (47.8%)	1 (4.3%)		
Spitting	Seniors	4 (16.7%)	3 (12.5%)	2 (8.3%)	13(54.2%)	2 (8.3%)	7.06	0.13
	Juniors	1 (4.3%)	7 (30.4%)	5 (21.7%)	10 (43.5%)	0 (0%)		
Laxative	Seniors	0 (0%)	1 (4.2%)	1 (4.2%)	17 (70.8%)	5 (20.8%)	4.81	0.19
	Juniors	0 (0%)	2 (8.7%)	4 (17.4%)	16 (69.6%)	1 (4.3%)		
Diuretics	Seniors	0 (0%)	3 (12.5%)	3 (12.5%)	14 (58.3%)	4 (16.7%)	9.24	0.06
	Juniors	1 (4.3%)	2 (8.7%)	0 (0%)	20 (87%)	0 (0%)		
Diet pills	Seniors	1 (4.2%)	1 (4.2%)	1 (4.2%)	18 (75%)	3 (12.5%)	3.34	0.50
	Juniors	1 (4.3%)	2 (8.7%)	1 (4.3%)	19 (82.6%)	0 (0%)		
Vomiting	Seniors	0 (0%)	1 (4.2%)	2 (8.3%)	18 (75%)	3 (12.5%)	1.67	0.64
	Juniors	0 (0%)	2 (8.7%)	1 (4.3%)	19 (82.6%)	1 (4.3%)		

Legend: χ2—chi-square test; *p*—probability.

## Data Availability

The data presented in this study are available on request from the corresponding author.

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
