# Peer review of "Principles of Rapid Weight Loss in Female Sambo Athletes"

_ijerph, 2021, doi:10.3390/ijerph182111356_

Round 1

Reviewer 1 Report

The manuscript analyses the different methods of RWL in SAMBO athletes. As a result of the research it is concluded that it is important to give reliable information to the athletes to whom the athletes turn to the most. Nutritional education to the group of coaches and athletes is of vital importance.

ORGANISATION AND EXPECTED COMPONENTS

The manuscript is well organised and structured. The paper submitted meets all the formal requirements and is of sufficient scientific quality to be accepted for publication.

The manuscript has all the components expected for a scientific publication. The sections are clearly defined and developed

It is an up-to-date paper in terms of the knowledge available in the field of study so far, reflected in a very updated bibliography, has a high scientific quality, is very well elaborated and has been structured very well in the report.

The objectives and questions set out in this paper are clearly defined and have been resolved one by one throughout the work. In order to reach these objectives, an appropriate and currently valid methodology has been used at all times, which is explained clearly and in detail in the text. In this way, it has been possible to draw conclusions from the results, which are relevant to the area of work in which this manuscript is framed. The initial theory is linked to the data obtained in the study. All the data necessary to justify the conclusions reached are presented. The results obtained are presented in an appropriate way, they are comprehensible, useful and significant for further research in the field. The discussion is well elaborated and the innovative nature of the results is clear. The conclusions are written in a clear and succinct way, being coherent with the objectives set and the results obtained.

In conclusion, the manuscript is well written, easy to understand for the readers, and the results are hopeful for further application. Therefore, it is recommended that this article qualifies for publication in the journal.

ETHICAL CONSIDERATIONS

The project number and Ethics Committee approval document are requested.

ADEQUACY OF THE ARTICLE FOR THE JOURNAL, HARD FACTS

The article is considered suitable for publication in the journal. It is an article that can benefit from the results obtained and advance the field of current knowledge.

STRENGTHS AND WEAKNESSES

As a weakness of the study, although the number of subjects is the optimal, I would like to know what method has been used to calculate the sample size, as well as the data used.

As a strength of the study, it is worth mentioning the originality of the study in a sport such as SAMBO, which is widely practised..

MINOR CRITICISM

As minor criticisms:

  1. I would consider it appropriate to specify which inclusion and exclusion criteria have been taken into account
  2. The project number and Ethics Committee approval document are requested.
  3. The method used to calculate the sample size and the data used are requested.

Author Response

  1. I would consider it appropriate to specify which inclusion and exclusion criteria have been taken into account.

There were no specific inclusion criteria, except ones outlined in the methods. We excluded all of the male participants and female participants younger than 18 years of age. Also, add line 78-79 to full-field inclusion criteria.

  1. The project number and Ethics Committee approval document are requested.

In addition to this text, we provide the document of the project number and ethical approval.

  1. The method used to calculate the sample size and the data used are requested.

The sample size was not calculated for this study because this has only an explorative purpose. If the reviewer considers that this represents a limitation to this study, it can be added in the study limitation section (lines 200-212)

Reviewer 2 Report

The topic of the paper is interesting. However such subjects, cannot be investigated with questionnaires only. It need an experimental approach with some physiological measurement, such as fluid loss, accurate body weight measurement, time-course of weight loss. Questionnaire can give a general view, but are not enough to draw sound scientific conclusions about strategies used by athletes. The conlusion part is quite poor and few informative.  Used statistics are not appropriate for the kind of data, no results fro Kolomogorov-Smirnof is provided, any information on statistical power is provided. The paper need a deep revision of english language-

Below soem detailed observation:

In line 55 is written "widely discusse in literature" and only 1 paper is cited as support of this statement.

line 58, must be database no date-base.

68 questionnaires not questioner , this mistake is all along the text.

71-72 must be specified "height and weight"

128-146-147 methods and influence, is not clear what is influence

Author Response

  1. The topic of the paper is interesting. However, such subjects, cannot be investigated with questionnaires only. It need an experimental approach with some physiological measurement, such as fluid loss, accurate body weight measurement, time-course of weight loss. Questionnaire can give a general view, but are not enough to draw sound scientific conclusions about strategies used by athletes. The conlusion part is quite poor and few informative. Used statistics are not appropriate for the kind of data, no results fro Kolomogorov-Smirnof is provided, any information on statistical power is provided. The paper need a deep revision of english language-

Thank you for your comment. We outline the study limitations in lines 196-208 concerning our experimental approach. We used the Kolmogorov-Smirnov test only to evaluate normal distribution among the sample, as a method to determine which statistical analyses we should use (as it can be seen in Test of normality). Since the data were not evenly distributed, we conducted a chi-square crosstabulations test. The paper was given to a lecturer for deeper English revision, as it can be seen in the document by track-changes. We also expanded/changed the conclusion. 

  1. In line 55 is written "widely discusse in literature" and only 1 paper is cited as support of this statement.

In line 59 we add two more review studies that should expand our research.

  1. line 58, must be database no date-base.

Regarding database, our group of researchers conducted the investigation on World Sambo championship held in Novi Sad. We use that database for conducting this investigation, but only focusing on female athletes. We had study published in Nutrients earlier DOI: 10.3390/nu13041063, only investigating male subjects.

  1. 68 questionnaires not questioner, this mistake is all along the text.

Corrected

  1. 71-72 must be specified "height and weight"

Corrected

  1. 128-146-147 methods and influence, is not clear what is influence

We add influence part in lines 148-149

Reviewer 3 Report

The aim of the manuscript was to determine the rapid weight loss procedures in combat sports (i.e., SAMBO athletes). The results are interesting and provide some additional and extended knowledge that can be used to better understand the effects linked to a rapid weight loss. 

However, I suggest the authors to add more information within the introduction, methods, and discussion sections to improve the readability of the manuscript.

Here below some specific comments:

Specific comments

Intro

I would suggest the authors to first focus on the issue related to RWL and then to present the SAMBO discipline. In my opinion, the readability of the text would benefit from this new text order. 

Lines 57-63: please add the study hypothesis to complete the logical structure of the introduction section.

M&M

line 95: connected? Replace it as apparently not appropriate.

line 99: remove comma after "determining"

Discussion

Discuss whether the study hypothesis has been verified.

lines 133-135: please be specific. Why?

lines 150-153: revise the English language, poor structure.

lines 161-165: the authors are invited to expand the cognitive aspects of RWL perhaps also in the introduction section.

What about study limitations? Perhaps referring to the methods (and associated variables) used to establish a rapid weight loss. What about the measure of body composition changes? 

Please consider these articles to also expand the study limitations 

Bioimpedance Vector References Need to Be Period-Specific for Assessing Body Composition and Cellular Health in Elite Soccer Players: A Brief Report

https://doi.org/10.3390/jfmk5040073

Author Response

Intro

I would suggest the authors to first focus on the issue related to RWL and then to present the SAMBO discipline. In my opinion, the readability of the text would benefit from this new text order.

Lines 57-63: please add the study hypothesis to complete the logical structure of the introduction section.

Thank you for your comments and suggestions. Regarding the introduction construction, we are aware of the improved readability that can be achieved with suggested changes, but in consultations with other researchers in the team, we agree that the current structure of the introduction is most appropriate for this research.  Also, we expanded the introduction and added a section concerning cognitive and mental health. Also, we add study hypothesis lines 66-69

Discuss whether the study hypothesis has been verified.

We discussed hypothesis; lines 142-148

lines 133-135: please be specific. Why?

Further explanation add line 144-146

lines 150-153: revise the English language, poor structure.

Corrected

lines 161-165: the authors are invited to expand the cognitive aspects of RWL perhaps also in the introduction section.

We expanded the introduction; lines 53-57

What about study limitations? Perhaps referring to the methods (and associated variables) used to establish a rapid weight loss. What about the measure of body composition changes?

Limitation section added; lines 196-208

Bioimpedance Vector References Need to Be Period-Specific for Assessing Body Composition and Cellular Health in Elite Soccer Players: A Brief Report

We included suggested reference  https://doi.org/10.3390/jfmk5040073

Reviewer 4 Report

General comment 

The present study aimed to investigate the effects of a rapid weight loss in female sambo athletes. The authors provided further notions for coaches and practitioners regarding the importance to take into account that the practice of reducing weight before sambo competition is a procedure conducted also in sambo athletes.

The manuscript is well written, the rationale is well established and the literature on the topic is cited appropriately. I believe that present article should be shared with the scientific community, but there are some points that should be addressed. I hope that my comments will be useful to improve the scientific quality of the manuscript.

Introduction:

Lines 54-60: it should be better to consider to implement the cited studies (ref. 13,14) regarding rapid weight loss in sambo athletes because it’s the topic of this paper.

Material and methods:

Lines 94: did the Authors provided further analysis regarding between-group comparisons (senior Vs. junior)?

Discussion:

Lines 153-159: Please consider to explore a paragraph regarding practical applications that coaches or practitioners should follow to avoid a rapid weight loss within their athletes.

Author Response

Lines 54-60: it should be better to consider to implement the cited studies (ref. 13,14) regarding rapid weight loss in sambo athletes because it’s the topic of this paper.

Thank you for suggestion. Regarding the additional explanation for citation (13,14) in the introduction, we address these studies later in the discussion (lines 156-158). We find that if we emphasize these two studies in the introduction, that would disturb the introduction's construct. However, if the reviewer insists on discussing these studies in detail, we can change this introductory section.

Lines 94: did the Authors provided further analysis regarding between-group comparisons (senior Vs. junior)?

Regarding the further analysis, we did not find any significant differences among seniors and juniors, and we did not conduct further analyses. The main reason for this kind of conclusion can be the age that participants start their RWL practicing (approximately 15 years of age) displayed in line 145-156

Lines 153-159: Please consider to explore a paragraph regarding practical applications that coaches or practitioners should follow to avoid a rapid weight loss within their athletes.

Line 170-174, we explore the practical application.

Round 2

Reviewer 2 Report

Albeit some effort has been made to improve the paper, the paper still lack of some fundamental aspects to be publshised.

English language is still poor and in some part not understandable:

line 17 : are limited

18: aimed at

58: please explain what ia "affective". What does it means.

86 probably do you mean flow chart of subject´s selection or procedure of subject selection ?

104 You mention the questonnaire was translated in several language. However, when a questionnaire, originally validated in one language is translated, it need to be validated again for that language. Thsi means all the questionnaires answered in language different form the original questionnaire are not reliable. This is a major flaw.

132 show the smaller influence not "had less than influence

158 share their experience with who ?

188 where do you find this information to lost 5% of body weight is harmful ? please cite a reference.

216. what mean championships were organized in the bubble ??

overall the paper can´t be accepted in the present form. Also the data obtained are very poor and poor informative.

Author Response

We are thankful for reviewer comments. We tried to answer or correct every suggestion that reviewer had outline.

line 17: are limited

Corrected

18: aimed at

Corrected

58: please explain what ia "affective". What does it means.

There is few definition of term affective, but in our context it is referred to:

  1. (expressive meaning) The personal feelings expressed by athlete.
  2. (attitudinal meaning) The personal feelings, attitudes, or values of an athlete inferred from their words and/or nonverbal behavior.

86 probably do you mean flow chart of subject´s selection or procedure of subject selection ?

Corrected

104 You mention the questonnaire was translated in several language. However, when a questionnaire, originally validated in one language is translated, it need to be validated again for that language. Thsi means all the questionnaires answered in language different form the original questionnaire are not reliable. This is a major flaw.

Thank you for your opinion. Questionnaires should likely be validated again for the other languages, primarily if the questionnaires assess psychological behavior. The validity and reliability of measuring instruments (questionnaire) are crucial in research because it ensures the effectiveness of testing hypotheses. However, in this case, RWL questionnaire used in this study had informative nature, and we think that assessing these questionnaires in several different languages did not influence validity of their answers.

132 show the smaller influence not "had less than influence

Corrected

158 share their experience with who ?

Corrected

188 where do you find this information to lost 5% of body weight is harmful ? please cite a reference.

Line is corrected and the reference are cite. Thank you for point on this issue.

  1. what mean championships were organized in the bubble ??

We expend the line with the further description of “bubble”. In addition to this letter we provide a Health measures during the world sambo championships 2020.

overall the paper can´t be accepted in the present form. Also the data obtained are very poor and poor informative.

We are thankful for your comments. We tried to answer or correct every suggestion that you outlined.